# Pathophysiology of Pulmonary Arterial Hypertension: Focus on Vascular Endothelium as a Potential Therapeutic Target

**DOI:** 10.3390/ijms26199631

**Published:** 2025-10-02

**Authors:** Michele Correale, Valentina Mercurio, Ester Maria Lucia Bevere, Beatrice Pezzuto, Lucia Tricarico, Umberto Attanasio, Angela Raucci, Anne Lise Ferrara, Stefania Loffredo, Claudio Puteo, Massimo Iacoviello, Maurizio Margaglione, Natale Daniele Brunetti, Carlo Gabriele Tocchetti, Piergiuseppe Agostoni, Claudio Mussolino, Maria Cristina Vinci

**Affiliations:** 1Cardiothoracic Department, ‘Policlinico Riuniti’ University Hospital, 71100 Foggia, Italy; lucia.tricarico.lt@gmail.com (L.T.); dott.claudioputeo@gmail.com (C.P.); massimo.iacoviello@gmail.com (M.I.); natale.brunetti@unifg.it (N.D.B.); 2Department of Translational Medical Sciences, University of Naples Federico II, Via Sergio Pansini 5, 80131 Naples, Italy; valemercurio@yahoo.com (V.M.); anneliseferrara@gmail.com (A.L.F.); stefanialoffredo@hotmail.com (S.L.); cgtocchetti@gmail.com (C.G.T.); 3Department of Medical and Surgicl Sciences, University of Foggia, 71100 Foggia, Italy; estermarialuciabevere@gmail.com; 4Centro Cardiologico Monzino, IRCCS, 20138 Milan, Italy; beatrice.pezzuto@gmail.com (B.P.); piergiuseppe.agostoni@cardiologicomonzino.it (P.A.); 5Department of Clinical Medicine and Surgery, Federico II University, 80131 Naples, Italy; umberto.attanasio@yahoo.it; 6Unit of Experimental Cardio-Oncology and Cardiovascular Aging, Centro Cardiologico Monzino IRCCS, 20138 Milan, Italy; angela.raucci@cardiologicomonzino.it; 7Center for Basic and Clinical Immunology Research (CISI), WAO Center of Excellence, University of Naples Federico II, 80131 Naples, Italy; 8Institute of Experimental Endocrinology and Oncology, National Research Council (CNR), 80131 Naples, Italy; 9Medical Genetics, Department of Clinical and Experimental Medicine, University of Foggia, 71122 Foggia, Italy; maurizio.margaglione@unifg.it; 10Department of Clinical Sciences and Community Health, Cardiovascular Section, University of Milan, 20142 Milan, Italy; 11Institute for Transfusion Medicine and Gene Therapy at Center for Translational Cell Research (ZTZ), University of Freiburg, 79085 Freiburg, Germany; claudio.mussolino@uniklinik-freiburg.de; 12Unit of Cardiovascular Epigenetics, Centro Cardiologico Monzino IRCCS, 20138 Milano, Italy; cristina.vinci@cardiologicomonzino.it

**Keywords:** pulmonary hypertension, pulmonary arterial hypertension, pulmonary circulation, pulmonary vascular disease, endothelium

## Abstract

Pulmonary arterial hypertension (PAH) is a rare condition characterized by high pulmonary artery pressure leading to right ventricular dysfunction and potential life-threatening consequences. It primarily affects the pre-capillary pulmonary vascular system. The exact pathophysiological mechanisms underlying PAH are not entirely known. Environmental factors; genetic predisposition; mitochondrial and microRNA dysfunction; and inflammatory, metabolic, and hormonal mechanisms may be involved. A central role is played by the dysfunction of the pulmonary vascular endothelium. This alteration is characterized by a reduction in vasodilatory and antiproliferative factors such as prostacyclin and nitric oxide and an increase in vasoconstrictive and mitogenic substances such as endothelin and thromboxane A2. Such imbalance leads to a progressive increase in pulmonary vascular resistance. The aim of the present review is to focus on the vascular endothelium and its role as a potential therapeutic target in PAH.

## 1. Introduction

Pulmonary hypertension (PH) is a pathophysiological condition characterized by an increase in mean pulmonary artery pressure (mPAP) of ≥20 mmHg in the pulmonary circulation measured by means of right heart catheterization. This elevation may be accompanied by (i) an increase in pulmonary artery wedge pressure (PAWP) > 15 mmHg defining the post-capillary form of PH, or (ii) an increase in pulmonary vascular resistance (PVR) > 2 Wood Units (WU) defining the pre-capillary form, or (iii) both conditions defining the combined post-capillary form [1,2,3,4].

Hemodynamic alterations from PH lead to an increase in right ventricular afterload, which results in right heart dysfunction and death if not adequately treated [5].

Pulmonary hypertension affects 1% of the world’s population [6]. According to the 2022 ESC/ERS guideline classification, it is categorized into five groups. Group 1 represents pulmonary arterial hypertension (PAH) which includes idiopathic forms, drug- and toxin-induced forms, connective tissue diseases, congenital heart defects, HIV infection, portal hypertension, and venous/capillary involvement forms. Group 2 comprises PH associated with left heart disease. Group 3 encompasses all forms associated with lung diseases and/or hypoxemia. Group 4 includes PH due to chronic pulmonary artery occlusion. Group 5 encompasses a wide variety of heterogeneous conditions, including hematologic and metabolic disorders. While those forms belonging to groups 2 and 3 are the most common, group 1’s PAH is a rare condition that primarily affects the pre-capillary pulmonary vascular system. The exact pathophysiological mechanisms underlying PAH are still debated. Environmental factors; genetic predisposition; mitochondrial and microRNA dysfunction; and inflammatory, metabolic, and hormonal mechanisms may be involved [1,7].

Over recent years, the role of pulmonary vascular endothelium in the pathophysiology of PAH has emerged. Indeed, PAH is characterized by a reduction in vasodilatory and antiproliferative factors such as prostacyclin and nitric oxide (NO) and an increase in vasoconstrictive and mitogenic substances such as endothelin and thromboxane A2. This imbalance leads to a progressive increase in pulmonary arterial resistance [1,2,3,8].

An increasing number of studies are focusing on the vascular endothelium and its role as a potential therapeutic target in PAH.

## 2. Pathophysiology of Pulmonary Hypertension

A central characteristic shared by all different types of PH is represented by the progressive pulmonary vascular remodeling, a process involving a range of structural and functional alterations predominantly affecting the distal part of pulmonary circulation. In PAH, such modifications primarily impact the pulmonary arterioles, whereas in conditions like PH associated with left heart disease, pulmonary veno-occlusive disease, or pulmonary capillary hemangiomatosis, changes are more pronounced in small-to-medium-sized veins and capillaries. Extensive involvement of the pulmonary circulation may translate to elevated PVR, consequently increasing pulmonary artery pressure, and ultimately leading to a relevant impairment in the function and the structure of the right-side chambers of the heart [9].

A fundamental feature of pulmonary vascular remodeling is the accumulation of vascular and immune cells within and around the vessel walls [10,11,12,13,14]. This process is characterized by changes in the intima, media, and adventitia of pulmonary arteries, and is driven by the proliferation and phenotypic transformation of cells. Factors that contribute to the development of PH include high shear stress, chronic hypoxia, genetic predispositions, and dysregulation of the TGF-β family (Figure 1). These factors can lead to vascular remodeling, endothelial dysfunction, and ultimately, increased PVR [15]. Alterations in the secretion of vasoconstrictors and vasodilators by pulmonary endothelial cells (ECs) play a pivotal role, alongside imbalances in factors regulating the pulmonary artery (PA), smooth muscle cells (SMCs), and fibroblast growth, as well as thrombotic and inflammatory mediators, and signaling pathways, such as the transforming growth factor (TGF)-β family impacting bone morphogenetic proteins (BMPs)—small mothers against decapentaplegic (Smad) 1/5/8 and activin-Smad2/3 pathways.

In idiopathic PAH (IPAH), distinct lesions (plexiform, obliterative, intimal medial hypertrophy, and adventitial remodeling) exhibit significant molecular heterogeneity, especially in plexiform and adventitial lesions, which show the highest number of differentially expressed genes [10,14]. These lesions are enriched for genes associated with or mutated in IPAH, particularly those related to BMP/TGF-β signaling, extracellular matrix, and endothelial–mesenchymal transition [16]. Furthermore, it has now emerged that vascular rarefaction in PAH, which has also been observed to varying degrees in other PH types, probably contributes to the elevation in PVR [17,18,19].

It is well-known that numerous gene mutations may increase the risk of developing PAH, especially those encoding TGF-β superfamily members [20,21]. Since understanding the complexities of these genes and their association with PAH remains extremely intricate, research is now focusing on identifying genetic modifiers that might offer protection or compensate for known mutations [22,23]. Alongside genetics, environmental factors, including hemodynamic forces, shear stress, drugs, toxins, hormones, inflammation, oxidative stress, and aging of the pulmonary circulation, play a vital role in influencing pulmonary vascular remodeling [24,25,26] (Figure 1).

Another mechanism implicated in cell accumulation within arterial walls in PAH involves the dysregulation in the expression and interaction of various receptor tyrosine kinases (RTKs). In particular, disturbances in serine-threonine kinases, particularly those in the BMP/TGF-β family, significantly contribute to pulmonary vascular remodeling in PAH [27]. This includes loss of BMPR-II expression and an overabundance of other receptors, leading to increased phospho-Smad2/3 [28]. Dysregulation of ligands in the BMP/TGF-β pathway, such as increased activin A and FSTL3, has also been observed, with relevant therapeutic implications. Inhibiting the activin pathway with agents like ACTRIIA-Fc has shown preclinical efficacy. Sotatercept, a soluble ACTRIIA IgG-Fc fusion protein, has demonstrated promising results in phase 2 and 3 clinical trials [29,30,31].

Of note, ion channels like KCNK3/TASK-1 and chloride channels, including TMEM16A and CFTR, seem to be implicated in PAH progression [32,33,34,35]. Also, dysregulated cellular metabolism, particularly in acetyl-CoA generation driven by enzymes like ACLY and ALDH1A3, contributes to vascular remodeling. Excessive metabolism of glutamine and serine may also lead to vascular stiffening [36,37]. Epigenetic modifications, including alterations in histone acetylation and methylation, and DNA methylation, are significantly involved in PAH pathology; in fact, the use of inhibiting epigenetic modifiers, such as BET proteins and HDACs, has shown promising results in preclinical studies and early clinical trials [38,39]. RNA modifications, such as N6-methyladenosine, and the role of factors like eukaryotic initiation factor 5A (eIF5A) and certain long noncoding RNAs (lncRNAs) also contribute to pulmonary vascular remodeling [40,41,42,43,44].

Several genes, including *BMPR2*, *ENG*, *ACVRL1*, *CAV1*, *SMAD9*, and *GDF2*, are associated with PAH predisposition, although often with low penetrance [2,20,21,45]. In particular, mutations in *GDF2*, encoding BMP-9, affect its processing and secretion, leading to decreased BMP-9 levels, which plays complex roles in pulmonary ECs, by influencing BMPR-II expression, apoptosis, barrier function, and potentially EMT [39,40,41]. Genes like *KDR*, *TBX4*, and *SOX17* have also emerged as PAH-associated genes involved in pulmonary circulatory system development and maintenance [46]. Genome-wide association studies have identified SNPs in loci linked to PAH, including enhancers for SOX17 and within the *HLA-DPA1/DPB1* gene [47]. Long-read sequencing technology can detect a broader range of genetic variations, offering further insights into the genetic basis of PAH [48].

PAH has a higher prevalence in women, and the penetrance of PAH-related mutations is greater in females. However, males with PAH tend to have a less favorable prognosis. These differences are partly attributed to variations in right ventricular adaptation to afterload, with women often showing superior adaptation, in which estrogens appear to have a dual role in PAH, contributing to lung vascular remodeling locally while offering systemic protection and improving right ventricular adaptation [49,50,51]. Clinical trials targeting estrogen levels have not shown significant benefits [52]. Genes related to PAH, such as *SOX17*, have been shown to interact with estrogens, potentially contributing to sex differences [53]. Molecular agents derived from sex chromosomes may also play a role in phenotypic sex differences in PAH [54,55,56,57,58].

## 3. Vascular Endothelium

The vascular endothelium can be compared to an organ in the human body, capable of performing a wide range of cellular signaling and synthetic functions. It is also under mechanical stress due to the shear forces generated by blood flow. The endothelial layer plays a crucial role in regulating vascular tone locally. The pulmonary endothelium is a single layer of ECs that lines the interior surface of blood vessels within the lungs. It covers the entire inner surface of arteries, veins, and capillaries within the vascular system [59].

The surface of the ECs is covered by the glycocalyx that is composed of proteoglycans, glycoproteins, and glycolipids.

The pulmonary endothelium cells play a crucial role in maintaining vascular homeostasis which is essential for effective gas exchange and overall respiratory function. Beyond its primary duty of facilitating the smooth passage of blood flow, the pulmonary endothelium contributes significantly to different biological processes crucial for lung and cardiovascular health and for disease diagnosis and management. This function is performed by regulating vascular tone, vessel integrity, blood flow, and smooth muscle cell proliferation and ensuring proper gas exchange. In this context the glycocalyx is fundamental in endothelial function. It is precisely the glycocalyx that regulates vascular permeability and tone, inflammation, and shear stress by interacting with different types of proteins [60,61]. Glycocalyx malfunction may lead to shear stress that in turn results in increased NO production by activating endothelial NO synthase (eNOS) [62].

Endothelial functions can be summarized as follows (Table 1):Vascular tone regulation: ECs produce substances that modulate vessel diameter and tone, controlling the degree of dilation and constriction of blood vessels. Key molecules include NO and prostaglandins, which promote vasodilation, and endothelin-1 (ET-1), which causes vasoconstriction. This careful regulation of vascular tone ensures a normal blood pressure.Anti-inflammatory and antithrombotic properties: These cells help suppress inflammation and prevent blood clot formation by releasing anti-inflammatory and antithrombotic agents, such as prostacyclin and thromboresistant factors. ECs play a pivotal role in the inflammatory response by checking the passage of leukocytes into the lung tissue. These cells present adhesion molecules that facilitate the binding and transmigration of leukocytes during inflammation and infection.Angiogenesis: The pulmonary endothelium is involved in the process of angiogenesis. This is crucial for tissue repair and regeneration, as well as for adapting to changes in oxygen demand and blood flow.Barrier Function: The endothelium acts as a selective barrier, preventing potentially harmful substances from entering the bloodstream and allowing the passage of those necessary.Gas Exchange Facilitation: The endothelium ensures that oxygen and carbon dioxide can efficiently pass between the lungs and the blood. This process is crucial for maintaining the body’s oxygen supply and removing carbon dioxide.Cellular Signaling: The pulmonary endothelium is involved in a complex network of signaling pathways that regulate cell growth, differentiation, and responses to injury. This is particularly important in maintaining the structural integrity of the blood vessels and responding to pathological conditions [58,63].Endothelial function is regulated by different and numerous factors such as neurotransmitters, catecholamines, and endocrine factors [64]. In relation to the important functions that the pulmonary endothelium performs, its alteration and, in particular, the destruction of its glycocalyx seem to play an essential role in the etiopathogenesis of PAH [58,61,65,66].

## 4. Role of Endothelium in the Pathophysiology of Pulmonary Hypertension

PH is characterized by the presence of pulmonary vascular endothelial dysfunction, which disrupts the normal balance of vasoactive, proliferative, and thrombotic factors in the lung circulation. In healthy lungs, the endothelium maintains low pulmonary artery pressure by releasing vasodilators (e.g., NO and prostacyclin) and restraining vasoconstrictors (e.g., ET-1) while also preventing abnormal cell growth and thrombosis [67]. In PH, this homeostatic function is lost—the pulmonary endothelium becomes chronically dysfunctional, driving many of the disease’s hallmark features [8]. Endothelial dysfunction in PH leads to an imbalance between vasodilation and vasoconstriction, promoting excessive vasoconstriction, vascular smooth muscle proliferation, a pro-thrombotic state (microthrombosis), and fibrotic remodeling of the pulmonary arterioles [8]. In fact, endothelial injury and dysfunction are now recognized as central, initiating factors in PH pathobiology [8], setting the stage for progressive vascular obliteration and increased PVR.

### 4.1. Molecular Imbalances

A dysfunctional endothelium in PH exhibits a shift in the production of key mediators. Notably, there is overproduction of vasoconstrictors and mitogens and a deficiency in vasodilators. For example, ET-1, a potent vasoconstrictor and pro-proliferative peptide produced by ECs, is overexpressed in PH, while endothelial-derived NO and prostacyclin (PGI_2_) are markedly reduced [68]. This skewed profile favors continuous vasoconstriction and vascular wall thickening. Studies have shown that in idiopathic PAH, ET-1 levels are elevated (along with other endothelium-derived vasoconstrictors like thromboxane A_2_), whereas prostacyclin and NO bioavailability are reduced, impairing vasodilation [69,70]. Such an imbalance was evidenced by observations that PAH patients have high thromboxane (TXA_2_) metabolite levels and low prostacyclin metabolites, and even experimental hypoxic PH models show increased TXA_2_ activity concomitant with reduced PGI_2_ synthesis [8]. The net effect of these molecular changes is persistent vasoconstriction of pulmonary arteries and chronic proliferative signaling to the vessel wall, which together drive up pulmonary arterial pressure. Beyond the canonical NO, prostacyclin, and endothelin pathways, the injured endothelium in PH also releases excess growth factors and inflammatory mediators. ECs in a chronically “activated” state secrete mitogenic factors like vascular endothelial growth factor (VEGF), basic fibroblast growth factor (FGF-2), and chemokines (e.g., CXC ligand 12), which promote the aberrant proliferation of pulmonary artery SMCs and adventitial fibroblasts [8,71]. At the same time, endothelial production of antiproliferative and antithrombotic substances decreases. This pro-proliferative, pro-coagulant milieu initiated by endothelial dysfunction is a driving force for the pathological vascular remodeling seen in PH. Over time, small pulmonary arteries develop concentric intimal thickening, smooth muscle hypertrophy, and in situ thrombosis, largely due to signals (or lack of signals) from the abnormal endothelium [8]. The impairment of flow-mediated dilation (FMD) values was associated with PAH in Systemic Sclerosis (SSc) patients and correlates with greater microvascular damage evaluated by Nailfold Video-Capillaroscopy. An inverse relationship between VEGF, angiopoietin-2, VCAM-1 levels, and FMD was observed, but only VEGF and angiopoietin-2 were significantly higher in patients with PAH [72]. Early identification of endothelial dysfunction could aid in predicting right ventricular dysfunction and PAH in SSc patients [73].

### 4.2. Endothelial Injury and Inflammation

The stimuli that provoke endothelial dysfunction in PH can be different. Chronic hemodynamic stress (high flow or pressure), hypoxia, inflammation, and genetic mutations (such as those in *BMPR2, ALK1*, or other PAH-related genes) are all implicated as initiating factors that injure the pulmonary endothelium [70]. When the endothelium is injured, a vicious cycle ensues: there is loss of the normal endothelial barrier and its quiescent phenotype, and in attempting to heal, the endothelium enters a state of proliferative and inflammatory activation [74]. Thus, the term “endothelial dysfunction” in PAH encompasses not only a reduced vasodilator property, but also a phenotypic change in which ECs acquire pro-inflammatory and pro-growth behavior. They upregulate adhesion molecules and cytokines that recruit inflammatory cells to the vessel wall, contributing to the perivascular inflammatory infiltrates observed in PAH lungs. ECs may also undergo endothelial-to-mesenchymal transition (EndMT) in response to injury, losing their typical characteristics and beginning to express mesenchymal cell markers [75]. This further disrupts vascular homeostasis and promotes extracellular matrix deposition and stiffening of the vessel. Overall, the loss of normal endothelial anti-inflammatory, anticoagulant, and antiproliferative properties, and the gain of abnormal new functions, sets off a cascade of pathological changes in the pulmonary circulation. One consequence is EC apoptosis in segments of the vasculature, which can paradoxically coexist with hyperproliferation of ECs in other segments. This dysfunction is so pivotal that one paradigm of PAH pathogenesis is an “imbalance between EC injury and repair”, the inability of the endothelium to recover, leading to progressive vascular obliteration [70].

### 4.3. Plexiform Lesions (Angioproliferative Foci)

A striking histopathological feature of advanced PAH is the presence of plexiform lesions, which underscore the role of the endothelium in disease. Plexiform lesions are complex, glomerulus-like clumps of proliferating ECs that nearly occlude the lumen of small pulmonary arteries. They are considered a hallmark of severe PAH. While certain studies have localized these lesions to branching points, the overall distribution of these lesions along the pulmonary artery remains unclear. Notably, plexiform lesions are composed primarily of ECs in various states of activation [76]. The ECs within plexiform lesions exhibit signs of dysregulated growth and angiogenesis; for example, they often express high levels of VEGF receptors and other proliferation markers, indicating an “angioproliferative” process [76]. This has led researchers to characterize PAH as a disease with *cancer-like traits in the endothelium*: the localized overgrowth of ECs in plexiform lesions resembles neoplastic processes. In fact, earlier studies by Tuder et al. noted “exuberant EC growth” with associated inflammation in these lesions, further highlighting that endothelial aberrations are at the core of occlusive lesion formation [77]. The presence of plexiform lesions confirms that endothelial dysfunction in PAH is not merely a loss of normal function, but also a gain of proliferative, disordered growth behavior that physically obstructs vessels.

### 4.4. Preclinical Evidence—Endothelium as the Driver

The causal role of endothelial dysfunction in PH pathophysiology is strongly supported by preclinical models. A relevant number of animal studies demonstrated that targeted injury to the pulmonary endothelium is sufficient in inducing PH, reproducing the human disease features as described in the following models.

#### 4.4.1. Monocrotaline Toxicity

Monocrotaline (MCT) is a pyrrolizidine alkaloid that, when given to rats, causes an acute, selective injury to pulmonary artery ECs. This endothelial damage triggers a cascade of events leading to severe PH within a few weeks [78]. Notably, monocrotaline-injected rats develop endothelial dysfunction followed by inflammation, smooth muscle proliferation, and vascular remodeling, mirroring idiopathic PAH. The MCT model’s efficacy in inducing PH underscores that endothelial injury alone can initiate the disease, as the toxin primarily targets the endothelium (for instance, via calcium-sensing receptor activation on ECs) to set off vascular disease [78]. However, the MCT model does not reproduce the complex plexiform lesions that characterize human PAH, nor does it reflect the genetic and immunological heterogeneity of patients; this limits its translational relevance to early-stage endothelial injury [79].

#### 4.4.2. SU5416/Hypoxia (“Two-Hit”) Model

Perhaps the most illuminating experimental model is the combination of a vascular endothelial growth factor receptor blocker (SU5416) with chronic hypoxia in rats. SU5416 induces apoptosis of pulmonary microvascular ECs, especially when the survival signal (VEGF) is blocked under low-oxygen conditions. In this model, an initial wave of EC death is followed by the emergence of occlusive endothelial proliferative lesions, remarkably akin to human plexiform lesions [80]. The outcome is severe PH with obliterative arteriopathy in the rats. Importantly, researchers found that if endothelial apoptosis is experimentally inhibited (using a caspase inhibitor), the rats do not develop the proliferative lesions or PH despite SU5416 and hypoxia exposure [80]. This finding illustrates a crucial point: endothelial apoptosis and subsequent “rogue” proliferation are fundamental to disease pathogenesis. In other words, the imbalance between EC loss and uncontrolled endothelial repair/growth can directly cause PH in vivo, which aligns with the pathological observations in human PAH [80]. Nevertheless, while this model is more representative of human PAH than MCT, it remains partially reversible in rodents and does not fully capture the multifactorial complexity of human disease [81].

A recent murine model combining the high-fat diet (HFD) with L-NAME administration (a nitric oxide synthase inhibitor) has been developed to emulate pulmonary vascular remodeling in PH associated with HFpEF. Mice exposed to HFD + L-NAME develop pulmonary hypertension, right ventricular dysfunction, small-vessel muscularization, and lung inflammation, including myeloid-derived IL 1β upregulation. Depletion of macrophages or neutralization of IL 1β attenuates these vascular changes, highlighting inflammation-driven remodeling mechanisms [82,83].

However, although these models demonstrate that disrupted endothelial homeostasis not only precedes the development of PH but actively drives the disease process, they are not fully predictive of human pathology. This underscores the need to combine classical models with next-generation approaches such as organoids, induced pluripotent stem cell-derived systems, and organ-on-chip technologies to bridge the translational gap.

### 4.5. Therapeutic Implications

Recognizing the central role of the endothelium in PAH pathophysiology has led to the development of current and emerging therapies (Figure 2). All three major classes of approved PAH treatments target endothelial pathways: endothelin receptor antagonists (ERAs) counteract the excess ET-1 from ECs, and phosphodiesterase-5 inhibitors (PDE-5is) and soluble guanylate cyclase (sGCS) stimulators enhance the NO–cGMP pathway of endothelial origin, with prostacyclin analogs/secretion enhancers supplementing the deficient prostacyclin pathway instead [84,85,86]. These three therapeutic pathways owe their success to rectifying the endothelial imbalance, alleviating vasoconstriction, and inhibiting vascular proliferation/thrombosis. However, although these drugs have demonstrated clinical benefit in terms of hemodynamic improvement, exercise capacity, and survival, their ability to fully restore endothelial function remains limited. Endothelin receptor antagonists are highly effective against vasoconstriction but have little impact on endothelial regeneration. Prostacyclin analogs provide broader antiproliferative and anti-inflammatory effects, though they require complex delivery and carry adverse effects. PDE5 inhibitors and sGC stimulators are safe and well-tolerated, but they do not address the underlying endothelial molecular defects. Emerging therapies such as sotatercept (targeting the TGF-β superfamily) and anti-inflammatory biologics (e.g., IL-6 or TNF-α inhibitors) aim to correct endothelial dysregulation at its root [87,88]. Beyond these, experimental strategies aim to restore healthy endothelial function more broadly, for instance, by regenerating the endothelium (i.e., via stem/progenitor cell therapies) or correcting molecular abnormalities in ECs (such as *BMPR2* gene therapy) [89,90,91,92]. Recently, another class of drugs has been shown to be effective on PAH by acting on a new and different pathway. Sotatercept, the first-in-class activin signaling inhibitor, showed reduced proliferation of endothelial and SMCs in vitro. It may improve pulmonary blood flow through inhibiting cellular proliferation, promoting cellular death, and decreasing inflammation in vessel walls. The focus on the endothelium as a potential therapeutic target is thus a direct consequence of its pivotal role in PH pathophysiology. By repairing endothelial dysfunction, restoring its antiproliferative, anti-inflammatory, and antithrombotic capabilities, it may be possible to halt or even reverse the vascular remodeling in PH. In summary, the pulmonary vascular endothelium is both the culprit and the key in PH: its injury and dysregulation initiate and drive disease progression, while its normalization remains an appealing strategy to treat and possibly prevent this destructive condition.

## 5. Vascular Endothelium as Potential Therapeutic Target in PAH

Given the crucial role of endothelial dysfunction in the pathogenesis of PAH, the primary goal of targeted treatment is to modulate the key pathways essential for maintaining endothelial homeostasis that are disrupted in the disease.

Until a few years ago, only three of the numerous pathways implicated in the development of PAH have been extensively studied and recognized as viable targets for specific therapies: the prostacyclin (PGI_2_), nitric oxide (NO), and ET-1 pathways.

The main effect of these drugs is to reduce PVR by promoting pulmonary vasodilation, although these treatments have also demonstrated some beneficial antiremodeling effects, while research in recent years is focusing on achieving more consistent antiproliferative effects. Recently, between drugs acting on the TGF-β superfamily pathway, sotatercept was approved for the treatment of patients with PAH. The maximal medical therapy for PAH is now four-drug therapy.

The following illustrates how both established drugs and innovative molecules recently introduced or currently under trial evaluation for PAH-specific treatment can act on endothelial dysfunction.

## 6. Established Specific PAH Drugs

### 6.1. Drugs Acting on Prostacyclin Pathway

Epoprostenol, a synthetic prostacyclin, was the first drug to be approved for PAH treatment [93], followed by prostaglandin analogs such as Iloprost, Treprostinil, and Beraprost (Beraprost only in Japan and Korea).

Synthetic prostanoids have a well-established role in PAH treatment in their various available formulations, having demonstrated to improve WHO-defined functional classes, exercise capacity (six-min walking distance—6MWD), hemodynamics, and survival in affected patients [1]. Beyond vasodilation, they also target endothelial dysfunction through multiple mechanisms [94].

Prostacyclin and all stable PGI_2_ analogs mainly act by potently binding to the prostaglandin I_2_ (IP) receptor, which is coupled via G-protein Gs to adenylyl cyclase and cyclic adenosine monophosphate (cAMP) production [92]. This interaction induces vasorelaxation and also exerts antithrombotic and antiproliferative effects through the activation of protein kinase A.

Additionally, prostacyclins can interact with other prostaglandin receptors (like EP receptors), which might contribute to or modulate their therapeutic action, and with peroxisome proliferator-activated receptors (PPARs), highly expressed in ECs (ECs) [92]. PPARs promote vasorelaxation and may contribute to the blood pressure-lowering effects of PGI_2_ [95]. Moreover, several studies have shown that treatment with PGI_2_ and its analogs increases VEGF production [94], linked to not only cyclic AMP, but also PPAR activation [93]. VEGF enhances NO and PGI_2_ production in the ECs [95], stimulates endothelial progenitor cell proliferation, and promotes new blood vessel formation [94]. There is also evidence that a PPAR is a critical downstream target for bone morphogenetic protein receptor type 2 (BMPR2) in both ECs and SMCs, being involved in the regulation of pulmonary arterial endothelial cell (PAEC) survival, proliferation, and migration [96].

Lastly, PGI_2_ has also shown anti-inflammatory properties, as it acts by inhibiting the expression of selectins and adhesion molecules in endothelial and inflammatory cells [97], thereby reducing leukocyte–endothelium interactions during inflammation, and downregulating pro-inflammatory cytokines and chemokines [1].

In recent years, an oral highly selective non-protanoid IP agonist, Selexipag, has been developed for PAH treatment.

In a phase 3 placebo-controlled trial on 1156 PAH patients, Selexipag has shown a lower risk of mortality from all causes or PAH-related complications in treated patients vs. placebo [98].

Ralinepag is a novel oral prostacyclin analog with an extended terminal half-life of 24 h (in contrast, the active metabolite of selexipag (MRE-269) exhibited a sharp peak with a half-life of 9–10 h [99]) that limits fluctuations in plasma concentrations.

Slow release from the ralinepag extended-release (XR) formulation supports once-daily dosing for all concentrations, whereas, for selexipag, the plasma PK profile is consistent with a need for more frequent dosing [100].

Ralinepag was evaluated in a double-blind phase 2 clinical trial [101], and it was shown to significantly decrease PVR compared with placebo in PAH patients on mono or dual combination background therapy.

A phase 3 trial with the primary endpoint of time-to-first clinical worsening is currently running to further assess the efficacy and safety of Ralinepag in a large population of group I PAH patients [102].

### 6.2. Drugs Acting on Endothelin 1 Pathway

The ERAs (Bosentan, Ambrisentan, and Macitentan) act by blocking the interaction between ET-1, a strong vasoconstrictor polypeptide primarily produced by vascular ECs, and its receptors, ETA and ETB [103]. Specifically, ETA receptors are found on pulmonary SMCs, where they induce significant vasoconstriction and cell proliferation; by contrast, ETB receptors, mainly located on vascular ECs, induce vasodilation by promoting NO and PGI_2_ production, and contribute to the pulmonary clearance of circulating ET-1. While ETB receptors have protective roles in ECs, they are also present in the muscle cells of vascular walls, where they can induce effects similar to those of ETA receptors, such as vasoconstriction and cell proliferation [104].

ERAs are divided into two types, selective (Ambrisentan) and non-selective (Bosentan and Macitentan), based on their action on ET receptors: the selectivity in blocking ETA receptors should allow for the preservation of the benefits of ETB receptors, such as vasodilation, while limiting the negative effects associated with ETA receptors.

Bosentan has also been shown to inhibit ET1-induced IFNγ release from CD4+ T cells [105], demonstrating an impact of ERAs on the inflammatory processes involved in PAH pathogenesis.

Moreover, 4-week Bosentan treatment has led to a rapid and sustained improvement in endothelial function as an increase in brachial artery flow-mediated dilation percentage (FMD%) in patients with SSc [106], known to be at high risk of developing PAH.

ERA treatment in PAH has demonstrated efficacy for symptoms, exercise capacity, hemodynamics, and time to clinical worsening [107,108,109]. Furthermore, the SERAPHIN [109] and AMBITION [110] trials on Macitentan and Ambrisentan, respectively, have shown the efficacy of upfront combination therapy of ERAs with PDE-5i.

Recently, a novel biological treatment specifically targeting ETA receptors was studied: Getagozumab, a murine antibody developed by hybridoma technology and then humanized as an IgG4 isotype, has been shown to significantly lower pulmonary arterial pressure in both hypoxia-induced and monocrotaline (MCT)-induced PAH monkey models and further reduce PAP and right ventricular hypertrophy in MCT-induced PAH monkeys [111], and a phase 1b trial is currently ongoing [112].

### 6.3. Drugs Acting on Nitric Oxide Pathway

The NO pathway can be addressed by two classes of PAH-specific drugs: the PDE-5i, including Sildanfil and Tadalafil, and the sGC stimulators, including Riociguat.

NO is an endogenous vasodilator produced by the pulmonary endothelium via endothelial NO synthase (eNOS). It activates guanylate cyclase, leading to the production of cyclic GMP (cGMP), which promotes the relaxation of SMCs and, consequently, vasodilation. Additionally, NO also regulates cellular proliferation, apoptosis, and inflammation, and is eventually degraded by PDE-5 [113].

Interestingly, Riociguat can stimulate soluble guanylate cyclase (sGC) independently of NO and also enhances sGC sensitivity to NO. Since PAH is often associated with reduced NO levels, Riociguat offers a valuable treatment option, particularly for patients not responding to PDE-5i [114].

Sildenafil and Riociguat demonstrated significant effects on EC dynamics, including decreased proliferation, increased apoptosis, and enhanced angiogenesis both in vitro and in vivo in patients with chronic thromboembolic pulmonary hypertension (CTEPH, group 4).

In clinics, Sildenafil, Tadalafil, and Riociguat have shown favorable effects on exercise capacity, symptoms, and/or haemodynamics [115,116,117,118,119]. Moreover, Tadalafil [118] and Riociguat also improved time to clinical worsening.

As oxidative stress associated with severe PAH leads to impairment in the NO/sGC signaling pathway, this shifts native sGC toward the unresponsive heme-free apo-soluble GS isoform and thus reduces the efficacy of NO pathway vasodilator drugs [119]. More recently the novel sGS activator Mosliciguat was generated, designed for local inhaled application in the lung and specifically activating apo-soluble GS [120]. In a phase 1 trial, inhaled Mosliciguat showed the same efficacy on PAP as inhaled Iloprost, Bosentan, or Sildenafil, without reduction in systemic blood pressure [121].

### 6.4. Drugs Acting on Transforming Growth Factor-Beta Superfamily Pathway

The TGF-β superfamily consists of two main signal transduction pathways: the TGF-β-activin-nodal pathway, acting via small mother against decapentaplegic (SMAD) 2/3 transcription factors, and the bone morphogenetic (BMP)-growth and differentiating factor (GDF) pathway, activating SMAD 1/5/8 [27,122].

In PAH, a dysregulation favoring SMAD 2/3 over SMAD 1/5/8 signaling, often due to BMPR2 gene mutations, may lead to endothelial dysfunction, SMC proliferation, and excessive extracellular matrix deposition [123]; thus, the correction of the imbalance between BMP and TGF-β signaling is a potential therapeutic target for PAH.

Sotatercept (ACTRIIA-Fc) is a soluble fusion protein composed of the extracellular domain of the human activin receptor type IIA linked to the Fc portion of human IgG1. By acting as a ligand trap for activin A/B and GDF 8/11, Sotatercept reduces activin-SMAD2/3 signaling, thus restoring SMAD signaling balance towards antiproliferative pathways [124]. In preclinical studies in murine PAH models, Sotatercept showed reduced proliferation of endothelial and SMCs in vitro and improvements in hemodynamics, RV hypertrophy, RV function, and arteriolar remodeling [123].

Sotatercept has been recently approved by the FDA (March 2024) and subsequently by the EMA (August 2024) for clinical use as the first-in-class activin signaling inhibitor for the treatment of PAH adult patients, thanks to the positive results of several randomized clinical trials.

The PULSAR study [30] was a phase 2, multicenter, randomized, double-blind trial that assessed the efficacy and safety of Sotatercept in 106 group 1 PAH patients in WHO functional class II or III and on stable background therapy. It included a 24-week placebo-controlled phase followed by an 18-month active treatment extension. Results showed a significant reduction in PVR for the Sotatercept group compared to placebo (the primary endpoint of the study), with a stronger effect at the higher dose tested. Improvements in 6MWD and a significant reduction in NT-proBNP levels were also observed. The most common adverse events were thrombocytopenia and an increased hemoglobin level.

During the extension phase [31], patients who switched from placebo to active treatment showed a significant reduction in PVR similar to those who received active treatment from the start. Secondary endpoints, including improvements in 6MWD and WHO functional class, were also met.

In the phase 2 SPECTRA trial [125], Sotatercept improved peak oxygen uptake, 6MWD, and resting and peak exercise hemodynamics assessed through invasive cardiopulmonary exercise testing (CPET) after 24 weeks of treatment.

The phase 3 double-blind placebo-controlled STELLAR trial [29] was conducted on 324 PAH patients in WHO functional class II or III and on stable background PAH therapy, confirming PULSAR’s findings, showing significant improvements in 6MWD and PVR, as well as a longer time to clinical worsening or death for the Sotatercept group compared to placebo. Common side effects were epistaxis, telangiectasia, dizziness, and an increase in hemoglobin by more than 2.0 g/dL in 12.3% of treated patients.

Further ongoing trials are evaluating Sotatercept’s effects on different parameters such as long-term safety and efficacy (SOTERIA), time to clinical worsening in newly diagnosed PAH patients (HYPERION) [126], time to first event for a combined endpoint in WHO III or IV functional class PAH patients (ZENITH) [127], efficacy, safety and tolerability in combined pre- and post-capillary PH due to HFpEF (CADENCE) [128], and on CPET parameters [129].

The addition of sotatercept to the treatment armamentarium led to significant optimism, as it is the first treatment to act on a completely novel pathway in nearly two decades, and four-pathway drug combination therapy is now a possible therapeutic scenario for patients with PAH.

## 7. Novel Specific PAH Drugs (Figure 2)

Several other molecules acting on TGF-β pathway are currently being evaluated as potential PAH-specific treatments.

Among these drugs, KER-012 is a fusion protein designed to specifically target activin A and B, GDF8, and GDF11, aiming to avoid side effects probably due to the lack of specificity seen with ligand traps like Sotatercept, such as increased hemoglobin and thrombocytopenia. In preclinical studies, KER-012 did not raise hemoglobin or red blood cells in primates [27], and the phase 2 trial TROPOS [130] is currently testing its safety and efficacy in adults with PAH on stable treatment after 24 weeks.

Another molecule acting on the TGF-β pathway is FK506 (Tacrolimus), an immunosuppressive drug mainly used in transplant patients, which acts via a dual mechanism of action as a calcineurin inhibitor that also binds FK-binding protein-12 (FKBP12), a repressor of BMP signaling; by releasing FKBP12 from type I receptors activin receptor-like kinase (ALK) 1, ALK2, and ALK3, Tacrolimus activates downstream SMAD1/5 [131]. Low-dose FK506 has been shown to reverse dysfunctional BMPR2 signaling in PAECs from idiopathic PAH patients, to prevent exaggerated chronic hypoxic PAH in mice with conditional BMPR2 deletion in ECs, and to reverse severe PAH in several rat models [130]. In a subsequent phase 2a trial [132] conducted in 23 PAH patients, Tacrolimus increased BMPR2 expression in peripheral blood mononuclear cells, but did not significantly improve 6MWD, echocardiographic parameters, or NTproBNP, suggesting that the effects of this drug in PAH patients should be tested in a larger clinical trial. Tacrolimus was generally well-tolerated, with nausea/diarrhea being the most commonly reported adverse event.

### 7.1. Drugs Acting on Inflammatory System

Chronic inflammation plays a key role in endothelial dysfunction and in the development of PAH. The inflammatory response of ECs can induce EC hyper-proliferation, endothelial barrier alterations, and vasoconstriction through activating multiple pathways and secretion of further mediators, finally accelerating pulmonary vascular remodeling and the progression of PH [74]. Increased levels of certain cytokines, especially IL-6, are linked to worse survival in patients with idiopathic and heritable PAH [133], and IL-6 blockade by the monoclonal anti-IL-6 receptor antibody MR16-1 has been shown to improve hypoxia-induced PAH in mice [134].

Tocilizumab, a humanized antibody against the IL-6 receptor, used in rheumatoid arthritis and juvenile arthritis treatment [135,136], has been tested in a phase 2 study on 19 patients with different types of PAH (excluding lupus, rheumatoid arthritis, and mixed connective tissue disease), but it did not significantly improve key clinical measures like PVR, 6MWD, or NT-proBNP levels [137]. An ongoing phase 2 trial, SATISFY-JP [138], is testing Satralizumab, another IL-6 receptor monoclonal antibody, in PAH patients with high IL-6 levels by assessing changes in PVR and 6MWD over 24 weeks.

In addition, tumor necrosis factor (TNF)-α inhibitors, already approved for conditions like rheumatoid arthritis and inflammatory bowel disease, might be promising for future PAH treatments. Indeed, in both smooth muscular and PAECs, TNF-α suppresses BMPR2 expression, which drives inappropriate cellular proliferation through dysregulated NOTCH2/3 signaling [139]. Etanercept, a TNF-α neutralizing agent consisting of soluble TNF receptor 2 conjugated to the Fc portion of human IgG, reversed PAH in rat models by restoring balance in BMP and NOTCH signaling [140].

Rituximab, a monoclonal antibody targeting B cells involved in autoimmune diseases like SSc and lupus, was tested in a phase 2 randomized trial in patients with SSc-associated PAH without severe interstitial lung disease [141], but failed to significantly improve 6MWD or PVR compared to the control group, while an ongoing single-center trial is evaluating Rituximab in 50 patients with lupus-associated PAH [142].

Finally, new therapies are exploring the use of cardiosphere-derived cells, which are heart-derived progenitor cells exhibiting multilineage potential and clonogenicity, and whose therapeutic effects are mainly achieved through the release of extracellular vesicles, such as exosomes [143]. Preclinical studies have demonstrated their anti-inflammatory and immunomodulatory effects [144], and preliminary findings from a phase 1a/b study on magnetic resonance parameters and 6MWD in PAH patients indicate encouraging results [145].

### 7.2. Drugs Acting on Growth Factors

Abnormal overproduction of growth factors like PDGF and VEGF, whose receptors belong to the RTK family, is believed to have a role in the excessive proliferation of PAECs and SMCs in PAH.

Imatinib, originally used for chronic myeloid leukemia and gastrointestinal stromal tumors due to its action on BCR-ABL and c-Kit, has shown benefits in animal models of PAH thanks to its effects in PDGFR inhibition. These findings led to the phase 3 IMPRES trial [146], where it improved 6MWD, PVR, and RV function. However, due to a high risk of bleeding complications, especially subdural hematomas in anticoagulated patients, Imatinib was eventually not approved for PAH treatment. The PIPAH trial is now investigating optimal dosing of Imatinib in PAH patients who are not on anticoagulation [147].

Conversely, inhaled Seralutinib, another PDGFR inhibitor, was well-tolerated and improved PVR in PAH patients in a phase 2 trial [148] and in the subsequent open-label extension study [149].

Phase 2b/3 and 3 trials of inhaled Imatinib [150] and Seralutinib [151] in PAH, respectively, are currently ongoing.

### 7.3. Drugs Acting on Metabolic Pathways

As PPAR-γ expression is reduced in PAH, both in murine models and patients [152], leading to abnormal cell proliferation, Rosiglitazone (PPAR-γ agonist) may contribute to PAH treatment by suppressing harmful signaling pathways in PAECs, like NOX4 expression, superoxide production, and PDGFR-β activation, as demonstrated in a murine hypoxia-induced PH model [153], and Pioglitazone restores fatty acid oxidation and mitochondrial homeostasis, improving right ventricular function in PAH models. However, this class of drugs must be used with caution in PAH patients due to potential side effects like fluid retention related to renal sodium reabsorption [154].

Metformin has shown promise in improving vascular dysfunction in PAH by enhancing NO production and suppressing cell proliferation in mice [155]. In a single-arm phase 2 trial [156], metformin therapy was safe and well-tolerated in 20 PAH patients, and was associated with improved RV fractional change at echocardiography, besides a reduction in RV triglyceride content that correlated with altered lipid and glucose metabolism markers. A larger phase 2 randomized clinical trial is currently running [138].

Other agents, such as SGLT2 inhibitors (SGLT2-is) Dapagliflozin [157,158,159], Empaglifloflozin, and Canagliflozin, and GLP-1 receptor agonist Liraglutide [160,161], have also demonstrated beneficial effects in PAH models, improving endothelial function and reducing inflammation. The therapeutic implications in PAH of these drugs are due to their metabolic effects, which include reducing aerobic glycolysis, improving mitochondrial function, and enhancing fatty acid oxidation. SGLT2-is reduce vascular inflammation and oxidative stress and reverse vascular dysfunction. SGLT2-is may decrease vascular remodeling and PASMC proliferation by inducing apoptosis [162,163]. Phase 2 randomized clinical trials are underway to evaluate the efficacy and safety of Dapagliflozin and Empagliflozin in PAH [138].

### 7.4. Drugs Acting on ncRNA

In recent years, increasing evidence has highlighted the role for noncoding ribonucleic acids (ncRNAs), including microRNA (miRNAs) and lncRNAs, in PAH pathogenesis. lncRNA modulates PASMC proliferation [164]. In particular, miRNAs have been shown to contribute to endothelial homeostasis. lncRNA may be involved in dysfunction of PAECs. lncRNA driven by super-enhancers HCG20 (HLA complex group 20) contributes to PAEC dysfunction through U2AF2 (U2 small nuclear RNA auxiliary factor 2)-mediated alternative splicing of EIF2AK2 (eukaryotic translation initiation factor 2 alpha kinase 2) [165].

Some, known as mechano-miRs, are sensitive to blood flow-induced shear stress in PAECs and regulate genes involved in cell function and growth [166]. Additionally, several miRNAs influence NO production [164].

Preliminary experimental studies have shown encouraging results: for example, the therapeutic supplementation of miR-181a-5p and miR-324-5p, which are lower in PAH patients, have been shown to reduce proliferative and angiogenic responses in patient-derived ECs and to attenuate disease progression in PAH mice [167]. Similarly, levels of miR-483, which targets several PAH-related genes, including TGF-β, TGF-β receptor 2 (TGFBR2), and ET-1, are reduced in PAH patients. However, its overexpression in murine MCT and a hypoxia-PH model has been shown to improve PH and reduce heart hypertrophy [168].

Although it represents an intriguing perspective, the clinical application of ncRNA-based therapies presents several challenges [164], such as the complex systemic delivery of ncRNAs, the need for repeated applications, the potential toxicity and interactions with commonly used drugs, and the absence of an antidote.Nevertheless, attempts have been made in recent years with interesting outcomes, although current trials are on miRNAs predominantly active in muscle vascular cells.Olaparib, a PARP-1 (a protein involved in the processes of DNA repair) inhibitor already used in cancer therapy [169], targets a DNA repair enzyme that is overactivated in PAH as a consequence of DNA damage due to chronic inflammation. The overactivation in turn reduces miRNA-204 levels [170] that promotes abnormal smooth muscle cell proliferation and survival by enhancing expression of bromodomain-containing protein 4 (BRD4) [171]. Olaparib is currently being evaluated in PAH patients within the OPTION phase 1b trial [172]. Additionally, Apabetalone, an oral BRD4 inhibitor, has shown preliminary promise in improving pulmonary vascular resistance and cardiac output when combined with standard PAH treatments [173].

## 8. Future Directions

Drug development for PAH in recent years is focusing on achieving more consistent antiproliferative effects, beyond aiming at pulmonary vasodilatation. Several pathways are being explored by researchers to achieve this goal. A few molecules, such as sotatercept, have been successfully tested in humans and have already received approval for clinical use in PAH. In the section “novel specific PAH drugs”, we reported other molecules acting on the TGF-β pathway currently evaluated as potential PAH-specific treatments. Other molecules acting on some growth factors’ pathways seem to be promising for the treatment of PAH (in particular, Seralutinib).

One might believe that research on drugs that act on the inflammatory system might be more advanced than other lines of research, given that some of these molecules (Tocilizumab) are already used to treat other diseases (rheumatoid arthritis and inflammatory bowel disease) and because inflammation is at the basis of many other pathological conditions (including heart failure and coronary artery disease, for example). However, to the best of our knowledge, these molecules are not approved for the treatment of PAH to date.

Other promising molecules in PAH research that can interact with the endothelium are drugs acting on metabolic pathways. It is very interesting what emerged from the Research Symposium on Pulmonary Injury and Repair of the Endothelium (ReSPIRE) [174]. The maladaptation of metabolic pathways in the lung endothelium may contribute to the progression of the PAH. Lipid metabolism is dysregulated in PAH patients and associated with PAH severity [175]. Thus, a role for fatty acid oxidation in promoting pulmonary microvascular EC dysfunction in PAH is under investigation [176,177].

These findings point to future research directions aimed at identifying mechanisms of dysregulated endothelial metabolism which could serve as therapeutic targets for pulmonary vascular diseases. However, the research on metabolites (e.g., BOHB) derived from fatty acid oxidation seems a long way from realizing a drug for PAH.

In the search for new drugs for PAH, much attention is being given to the inclusion of patients with SSc in studies because SSc-PAHs are the most frequently associated forms of PAH and are systematically screened for the possible development of PAH. However, for the forms associated with HIV infection, the development of specific drugs is more difficult, also due to possible pharmacological interactions with antiretroviral drugs.

In recent years, the role of echocardiography in phenotyping PAH patients and improving prognostic stratification has been emphasized [178,179]. Improvements in outcome were achieved when PAH-specific therapies reduced right ventricular size and improved right ventricular function.

Future directions should provide a comprehensive risk stratification incorporating right ventricular function and a mechanism-based treatment paradigm, encouraging a tailored therapeutic approach in PH management. One possible way forward would be the development of RV-targeted therapies.

Even though the foundations for precision medicine have been laid, we are still very far from its implementation in the management of PAH. Personalized therapy based on genomic, proteomic, and metabolomic profiling is only a promising avenue. Single-cell RNA sequencing (scRNA-seq) is identifying unique transcriptional changes in endothelial and PASMCs [180].

## 9. Conclusions

PAH is a rare condition characterized by high pulmonary artery pressure leading to right ventricular dysfunction and potential life-threatening consequences. In its pathogenesis, a central role is played by the dysfunction of the pulmonary vascular endothelium. This alteration is characterized by a reduction in vasodilatory and antiproliferative factors such as prostacyclin and NO and an increase in vasoconstrictive and mitogenic substances such as ET-1 and thromboxane A2. The vascular endothelium is a potential therapeutic target and, in the near-future, experimental research could be conducted to test new molecules to preserve or restore endothelial function in patients already affected by PAH or in patients at risk (e.g., with connective tissue diseases).

The knowledge of the physiology and pathophysiology of the endothelium in PAH is essential for the development of new and specific therapeutic targets for the treatment of PAH.

## Figures and Tables

**Figure 1 ijms-26-09631-f001:**
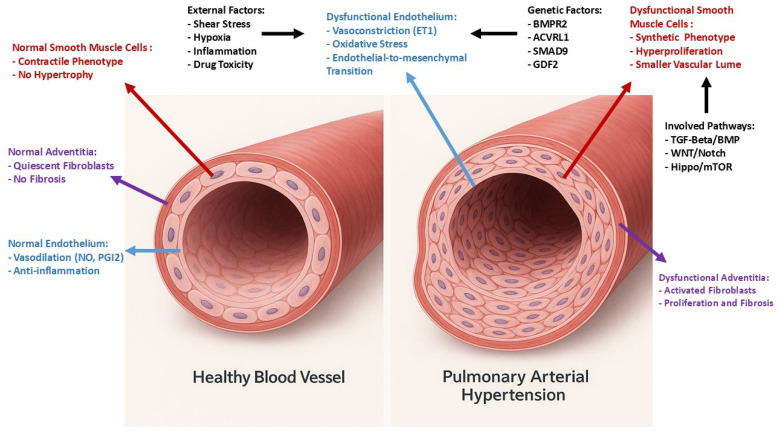
Role of endothelium in pathophysiology of pulmonary hypertension.

**Figure 2 ijms-26-09631-f002:**
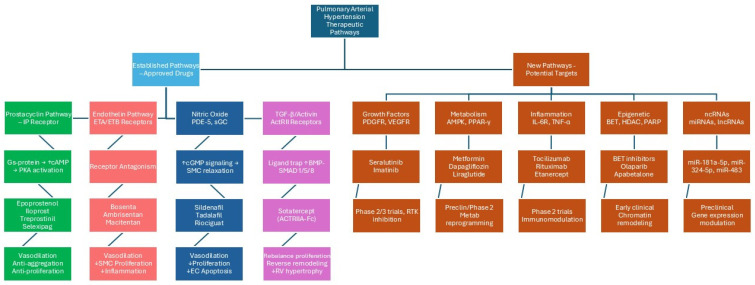
Pulmonary arterial hypertension therapeutic pathways (established pathways and new pathways/potential new targets).

**Table 1 ijms-26-09631-t001:** Function of pulmonary endothelium and associated mechanisms/key molecules.

Function	Description	Mechanisms/Key Molecules	Approved Therapeutic Agents	Experimental Therapeutic Agents
Vascular tone regulation [63]	Modulator of vessel diameter to maintain blood pressure and proper blood flow [63]	*NO* *Prostaglandins* *ET-1* [63]	*Sildenafil, Tadalafil, Bosentan, Ambrisentan, Macitentan, Riociguat, Epoprostenol, Iloprost, Treprostinil, Selexipag*	*Ralinepag, miRNAs*
Anti-inflammatory and antithrombotic properties [62,63]	Preventer of inflammation and thrombus formation, regulator of leukocyte migration into lung tissue [62,63]	*Prostacyclin* *Thromboprotective factors* *Adhesion molecules* (e.g., *ICAM-1*, *VCAM-1*) [62,63]		*Tocilizumab, Satralizumab, Etanercept, Dapagliflozin, Empagliflozin, Canagliflozin, Liraglutide*
Angiogenesis [62,63]	Promotes new vessel formation, essential for tissue repair and adaptation to oxygen demand changes [62,63]	*PDGF and VEGF* *Angiopoietin-2* *Other proangiogenic factors* [62,63]		*Imatinib, Seralutinib, miRNAs*
Barrier Function [12,13]	Acts as a selective barrier, preventing harmful substances from entering the bloodstream while allowing essential molecules to pass [62,63]	*Endothelium glycocalyx* *Tight junction proteins* [62,63]		
Gas Exchange Facilitation [63]	Ensures efficient oxygen and carbon dioxide transfer between alveoli and bloodstream [63]	*Alveolar-capillary endothelium* [63]		
Cellular Signaling [58,63,64]	Involved in signaling pathways regulating cell growth, differentiation, and responses to injury [58,63,64]	*Signaling cascade* (e.g., *MAPK, PI3K/AKT pathways*) *Regulatory molecules* [58,63,64]	Sotatercept	*KER-012, FK506 (Tacrolimus), Rosiglitazone, Imatinib, mTOR, Tamoxifen, Anastrazole*
Response to shear stress [11]	Detects blood flow changes, promotes NO production to preserve vascular homeostasis [60,61]	*eNOS* [62] *Endothelial glycocalyx* [60,61]		*Metformin, mechano-miRs, miRNAs*

Table footnote. NO: nitric oxide; ET-1: endothelin-1; VEGF: vascular endothelial growth factor; ICAM-1: Intercellular Adhesion Molecule 1; VCAM-1: Vascular Cell Adhesion Molecule 1; MPAK: Mitogen-Activated Protein Kinase; PI3K/AKT: Phosphoinositide 3-kinase/Protein kinase B; eNOS: endothelial NO synthase.

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
