# Peer review of "Pathophysiology of Pulmonary Arterial Hypertension: Focus on Vascular Endothelium as a Potential Therapeutic Target"

_ijms, 2025, doi:10.3390/ijms26199631_

Round 1

Reviewer 1 Report

Comments and Suggestions for Authors

The article by Correale et al is well written and informative, addressing the pathophysiology of pulmonary arterial hypertension. It is nice to read about the recent advances in the management of PAH. The reviewer has the following comments about the review.

L48-49: Hemodynamic alterations from PAH lead to an… (inserted ‘from PAH’)

L50: PH affects 1% of the world population (needs reference). Also, expand PH; it cannot begin the sentence with an abbreviation.

L59: The precise pathophysiological mechanism underlying PAH are not entirely known – delete this sentence. Replacement sentence: The pathophysiological mechanisms underlying PAH are being studied.

L63/L64: A central role in the….endothelium: delete; Replacement sentence: Pulmonary vascular endothelium plays a central role in the pathophysiology of PAH.

L71/L72: Delete the work ‘adverse’

L93: Expand EC for the first time. This has not been expanded before

L98/L99: Contributing factors for pulmonary vascular remodeling include high shear stress…….dysregulation of the proteins of the TGF-beta family. …. (underlines: new wording inserted).

The article needs to be reviewed for grammar and sentence structure.

  1. Pathophysiology of Pulmonary Hypertension.

The description could be made more engaging by a figure that explains the pathophysiology of PH.

Also, in the next section on vascular endothelium, the pathophysiology can be illustrated by another figure depicting the vascular endothelium and the vascular smooth muscle.

A figure illustrating the vascular endothelial targets in the management of PH could help in relating the pathophysiology to therapeutic targets, leading to better engagement.

The review is good. However, figures enhance the reader's engagement with the article. I would suggest that the authors be creative in coming up with figures, so that they are visually engaging.

Table 1: First line: Modulates of vessel diameter (spelling of the word vessel). It should read Modulator of vessel diameter OR Modulates vessel diameter…

Figure 1: This seems to be copied from another article. Needs reference or with permission etc. The reviewer suggests modifying this figure to describe the Vascular endothelium better as it relates to pathophysiology and therapeutic targets.

L438 – L442: The authors need to give the reader an idea why Ralinepag is better than Selexipag. What is the half-life of Selexipag? Do they differ significantly on PKPD.

L671. Again delete…The precise pathophysiological….not entirely known (delete). The authors have stated the same sentence twice and not needed. So, what is the purpose of writing a review?

Last sentence: Therefore, knowledge of the …..PAH. Delete the word ‘Therefore’. It should begin as ‘The knowledge…..This is a review, not a research article. You did not prove anything!

Can authors speculate on future directions in research and clinical therapeutic targets based on the review? What drugs will these patients benefit from, and is there a specific group of patients that would benefit from certain drugs? The authors have used terms like ‘tailored medicine’ and’ precision medicine’. Are we getting close to precision medicine? Or are we a long way from it?  All these can be put into a paragraph before the conclusions.

Comments on the Quality of English Language

None

Author Response

Reviewer 1: Inizio modulo

The article by Correale et al is well written and informative, addressing the pathophysiology of pulmonary arterial hypertension. It is nice to read about the recent advances in the management of PAH.

Thank you very much for your comment

The reviewer has the following comments about the review.

L48-49: Hemodynamic alterations from PAH lead to an… (inserted ‘from PAH’)

we thank the reviewer for this advice; we have now changed accordingly his/her suggestion

L50: PH affects 1% of the world population (needs reference). Also, expand PH; it cannot begin the sentence with an abbreviation.

we thank the reviewer for this advice; we have now changed accordingly his/her suggestion

L59: The precise pathophysiological mechanism underlying PAH are not entirely known – delete this sentence. Replacement sentence: The pathophysiological mechanisms underlying PAH are being studied.

we thank the reviewer for this advice; we have now changed accordingly his/her suggestion

L63/L64: A central role in the….endothelium: delete; Replacement sentence: Pulmonary vascular endothelium plays a central role in the pathophysiology of PAH.

we thank the reviewer for having pointed out this point; accordingly, we have replaced the sentence.

L71/L72: Delete the work ‘adverse’

we thank the reviewer for this advice; we have now changed accordingly his/her suggestion (delete the word “adverse”).

L93: Expand EC for the first time. This has not been expanded before

we thank the reviewer for having pointed out this point; accordingly, we have expanded EC.

L98/L99: Contributing factors for pulmonary vascular remodeling include high shear stress…….dysregulation of the proteins of the TGF-beta family. …. (underlines: new wording inserted).

we thank the reviewer for having pointed out this point; accordingly, we have now changed accordingly his/her suggestion

The article needs to be reviewed for grammar and sentence structure.

we thank the reviewer for having pointed out this point; The article was reviewed for grammar and sentence structure

  1. Pathophysiology of Pulmonary Hypertension.

The description could be made more engaging by a figure that explains the pathophysiology of PH.

we thank the reviewer for having pointed out this point; a figure (Fig. 1) that explains the pathophysiology of PH was inserted

Also, in the next section on vascular endothelium, the pathophysiology can be illustrated by another figure depicting the vascular endothelium and the vascular smooth muscle.

we thank the reviewer for having pointed out this point; a figure depicting the vascular endothelium and the vascular smooth muscle was inserted.

A figure illustrating the vascular endothelial targets in the management of PH could help in relating the pathophysiology to therapeutic targets, leading to better engagement.

we thank the reviewer for having pointed out this point; a figure (Fig 2)illustrating the pulmonary arterial hypertension therapeutic pathways. A Figure comprising both established pathways and new potential targets.

The review is good. However, figures enhance the reader's engagement with the article. I would suggest that the authors be creative in coming up with figures, so that they are visually engaging.

Thank you for your comments. We tried to be more creative, completely replacing the first version figure with a new, highly detailed figure.

Table 1: First line: Modulates of vessel diameter (spelling of the word vessel). It should read Modulator of vessel diameter OR Modulates vessel diameter…

we thank the reviewer for this advice; we have now changed accordingly his/her suggestion

Figure 1: This seems to be copied from another article. Needs reference or with permission etc. The reviewer suggests modifying this figure to describe the Vascular endothelium better as it relates to pathophysiology and therapeutic targets.

we thank the reviewer for this suggestion; the figure was changed accordingly his/her suggestion

L438 – L442: The authors need to give the reader an idea why Ralinepag is better than Selexipag. What is the half-life of Selexipag? Do they differ significantly on PKPD.

We reported in the text these data about ralinepag and selexipag.

L671. Again delete…The precise pathophysiological….not entirely known (delete). The authors have stated the same sentence twice and not needed. So, what is the purpose of writing a review?

we thank the reviewer for this advice; we have now deleted this sentence.

Last sentence: Therefore, knowledge of the …..PAH. Delete the word ‘Therefore’. It should begin as ‘The knowledge…..This is a review, not a research article. You did not prove anything!

we thank the reviewer for this advice; we have now changed accordingly his/her suggestion

Can authors speculate on future directions in research and clinical therapeutic targets based on the review? What drugs will these patients benefit from, and is there a specific group of patients that would benefit from certain drugs? The authors have used terms like ‘tailored medicine’ and’ precision medicine’. Are we getting close to precision medicine? Or are we a long way from it?  All these can be put into a paragraph before the conclusions.

we thank the reviewer for this advice; we have now added a paragraph about future directions before the conclusions, trying to answer all reviewer’s comments.

Comments on the Quality of English Language

None

Reviewer 2 Report

Comments and Suggestions for Authors

This review addresses an important and timely topic, focusing on the role of vascular endothelial dysfunction in pulmonary arterial hypertension (PAH) and its potential as a therapeutic target. The manuscript is generally well-structured, comprehensive, and demonstrates a good command of current literature, including both established therapies and emerging pharmacological strategies. The breadth of coverage, from pathophysiological mechanisms to preclinical and clinical therapeutic implications, is a clear strength. However, the manuscript would benefit from greater critical synthesis, improved clarity in certain sections, and a more consistent academic style. In its current form, it reads more as an extensive compilation of information rather than a critical, integrative review that advances the reader’s understanding.

Major Comments

  1. While the review promises a focus on the vascular endothelium, a substantial portion of the manuscript digresses into broader aspects of PAH pathophysiology without clearly linking these to endothelial mechanisms. This dilutes the central theme. Consider streamlining the introductory pathophysiology section to emphasize aspects directly relevant to endothelial biology and its therapeutic modulation.

  2. Some therapeutic subsections (e.g., drugs acting on metabolic pathways, ncRNA-based strategies) are informative but lack explicit discussion of how these interventions impact endothelial function, which is the manuscript’s stated focal point.

  3. The review tends to summarize published findings without critically appraising their strengths, limitations, or translational potential. For example, when describing preclinical models such as monocrotaline or SU5416/hypoxia, it would be useful to briefly discuss their limitations in replicating human PAH endothelial pathology.

    • In the therapeutic sections, adding a comparative perspective on efficacy, safety, and mechanistic relevance to endothelial restoration would significantly enhance the paper’s analytical value.

  4. The manuscript would benefit from a schematic or table explicitly linking specific endothelial dysfunction mechanisms (e.g., reduced NO bioavailability, increased ET-1 production, glycocalyx degradation) with corresponding approved and experimental therapeutic agents.

    • While three classical endothelial pathways (NO, PGIâ‚‚, ET-1) are well-described, their interplay with emerging targets (e.g., TGF-β superfamily modulation, anti-inflammatory approaches) should be better integrated into a cohesive therapeutic framework.

  5. Figures and Tables

    • Figure 1 is relevant, but additional high-quality, original figures illustrating the cascade from endothelial injury to vascular remodeling, and mapping therapeutic targets along this cascade, would improve reader comprehension.

    • Table 1 is useful, but could be reformatted for clarity and aligned with the manuscript’s therapeutic discussions.

Author Response

Reviewer 2:

Comments and Suggestions for Authors

This review addresses an important and timely topic, focusing on the role of vascular endothelial dysfunction in pulmonary arterial hypertension (PAH) and its potential as a therapeutic target. The manuscript is generally well-structured, comprehensive, and demonstrates a good command of current literature, including both established therapies and emerging pharmacological strategies. The breadth of coverage, from pathophysiological mechanisms to preclinical and clinical therapeutic implications, is a clear strength.

Thank you very much for your comment

However, the manuscript would benefit from greater critical synthesis, improved clarity in certain sections, and a more consistent academic style. In its current form, it reads more as an extensive compilation of information rather than a critical, integrative review that advances the reader’s understanding.

we thank the reviewer for this advice; we have now changed accordingly his/her suggestion, trying to offer a more critical approach in the future directions section.

Major Comments

  1. While the review promises a focus on the vascular endothelium, a substantial portion of the manuscript digresses into broader aspects of PAH pathophysiology without clearly linking these to endothelial mechanisms. This dilutes the central theme. Consider streamlining the introductory pathophysiology section to emphasize aspects directly relevant to endothelial biology and its therapeutic modulation.

we thank the reviewer for this advice; we have now changed accordingly his/her suggestion [simplifying the introductory section on pathophysiology (from 1500 words to 910) to emphasize aspects directly relevant to endothelial biology].

  1. Some therapeutic subsections (e.g., drugs acting on metabolic pathways, ncRNA-based strategies) are informative but lack explicit discussion of how these interventions impact endothelial function, which is the manuscript’s stated focal point.

we thank the reviewer for this advice; we have now discussed how future interventions may impact endothelial function

  1. The review tends to summarize published findings without critically appraising their strengths, limitations, or translational potential. For example, when describing preclinical models such as monocrotaline or SU5416/hypoxia, it would be useful to briefly discuss their limitations in replicating human PAH endothelial pathology.

The authors discussed the limitations of preclinical models as seggested by reviewer.

    • In the therapeutic sections, adding a comparative perspective on efficacy, safety, and mechanistic relevance to endothelial restoration would significantly enhance the paper’s analytical value.

The authors added a comparative perspective on efficacy, safety, and mechanistic relevance to endothelial restoration

  1. The manuscript would benefit from a schematic or table explicitly linking specific endothelial dysfunction mechanisms (e.g., reduced NO bioavailability, increased ET-1 production, glycocalyx degradation) with corresponding approved and experimental therapeutic agents.
  1. we thank the reviewer for this advice; we have now changed the table 1 accordingly his/her suggestion

    • While three classical endothelial pathways (NO, PGIâ‚‚, ET-1) are well-described, their interplay with emerging targets (e.g., TGF-β superfamily modulation, anti-inflammatory approaches) should be better integrated into a cohesive therapeutic framework.

we thank the reviewer for this advice; we have now changed this section accordingly his/her suggestion and furthermore, we moved the section where we talked about the sotatercept from the chapter on novel specific PAH drugs to the one where we discussed “Established specific PAH drugs” underlining how today the best therapy involves acting simultaneously on 4 pathways.

  1. Figures and Tables
    • Figure 1 is relevant, but additional high-quality, original figures illustrating the cascade from endothelial injury to vascular remodeling, and mapping therapeutic targets along this cascade, would improve reader comprehension.
    • we thank the reviewer for having pointed out this point; The authors changed the figure number 1, completely replacing the first version figure with a new, highly detailed figure

 Table 1 is useful, but could be reformatted for clarity and aligned with the manuscript’s therapeutic discussions.

we thank the reviewer for this advice; we have now changed the table 1 accordingly his/her suggestion.

Round 2

Reviewer 2 Report

Comments and Suggestions for Authors

I have no comments.